# Quantum tricriticality of incommensurate phase induced by quantum domain walls in frustrated Ising magnetism

Zheng Zhou,[1] Dong-Xu Liu,[2] Zheng Yan,[3, 1, *] Yan Chen,[1, 4, †] and Xue-Feng Zhang[2, ‡]

[1]*Department of Physics and State Key Laboratory of Surface Physics, Fudan University, Shanghai 200438, China*
[2]*Department of Physics, and Center of Quantum Materials and Devices,*
*Chongqing University, Chongqing, 401331, China*
[3]*Department of Physics, The University of Hong Kong, Pokfulam Road, Hong Kong, China*
[4]*Collaborative Innovation Center of Advanced Microstructures, Nanjing 210093, China*

Incommensurability plays a critical role not only in the strongly correlated systems such as frustrated quantum magnets. Meanwhile, the origin of such exotic order can be theoretically understood in the framework of quantum domain wall excitations. Here, we study the extended anisotropic transverse Ising model in the triangular lattice. Using the large scale quantum Monte Carlo simulations, we find that spatial anisotropy can stabilize the incommensurate phase out of the commensurate clock phase. Both the structure factor and the domain wall density exhibit the linear relationship between incommensurate order wave vector and number of domain walls, which is reminiscent of hole density in under-doped cuprate superconductors. Different from an indirect clue of the domain wall's existence, we find direct evidence from the features in the excitation spectrum. On the other hand, when introducing the next nearest neighbor interaction, we observed a novel quantum tricritical point related to the incommensurate phase. After carefully analyzing the energy of the ground state with different domain wall numbers, we conclude this tricriticality is non-trivial because it is caused by effective long-range inter-domain wall interactions with two competing terms $\frac{B}{r^\alpha} - \frac{C}{r^\gamma}$. At last, we focus on the clock phase, which is highly related to recently discovered material TmMgGaO$_4$, and find that the so-called "roton" mode results from the merging of domain wall mode and vortex-antivortex mode which breaks the local constraint of spin configuration.

## I. INTRODUCTION

The frustrated magnetism is the frontier of condensed matter physics, because the related materials can keep macroscopic degeneracy even at very low temperature [1, 2]. As a representative category, the quantum magnetic materials with triangular lattice geometry exhibit various exotic quantum phases, such as coplanar phase [3–5], spin density wave [6–8], spin glass [9] and so on [10]. Meanwhile, some of them are found to be disordered even when approaching zero temperature [11–13], so people believe that their ground states are quantum spin liquid [14–16] which holds symmetry enriched topology [17].

As the simplest model, the spin-1/2 in the triangular lattice with antiferromagnetic nearest-neighbor Ising interactions have to fulfill the triangle-rule or ice-rule in each triangle at low temperature, so its ground state is degenerate and has residue entropy [18–20]. When introducing the quantum fluctuation, such as antiferromagnetic spin exchange interaction[3–5] and transverse field [18–22], the degeneracy will be lifted, and the translational symmetry will be broken, so that the system enters an ordered phase. Such phenomena are called order from disorder [23]. However, in some cases, the quantum fluctuations are not enough to remove the degeneracy. As some examples, materials with spatial anisotropy [12, 24],

long-range interaction [25] or spin-orbital coupling [13] show many topological features, such as spinon dispersion in the energy spectrum and the *incommensurability* near the spin liquid phase.

The incommensurability or high-order commensurability is usually reminiscent of the famous high-$T_c$ cuprate superconductors [26, 27]. When Mott-insulator with antiferromagnetic spin order is doped with holes, the string type topological excitations will be constructed and divide the whole system into many domains, so that an incommensurate order is formed [28–31]. On the other hand, the ultra-cold atoms in the optical lattice used for simulating the Hubbard model also exhibit the string pattern excitations [32, 33], and the incommensurate supersolid phase is theoretically predicted for the repulsive hard-core bosons trapped in the anisotropic triangular optical lattice [34]. However, the incommensurate phase induced by the string type topological defects is still not fully understood, especially the related quantum phase transition and spectrum which is more experimentally feasible [31, 35, 36]

In this manuscript, we considered the transverse field Ising model (TFIM) in anisotropic triangular lattice, which is highly related to several strongly correlated systems, such as frustrated magnetism [37–39], antiferroelectric material [40] and trapped ions [41]. As shown in Fig. 1, the spatial anisotropy can turn the clock phase into the quantum incommensurate phase, which can be understood with the quantum topological defect – string type domain wall (DW) excitations. Meanwhile, the spectrum obtained with stochastic analytical continuation (SAC) [42, 43] demonstrates the low en-

* zhengyan@hku.hk
† yanchen99@fudan.edu.cn
‡ zhangxf@cqu.edu.cn

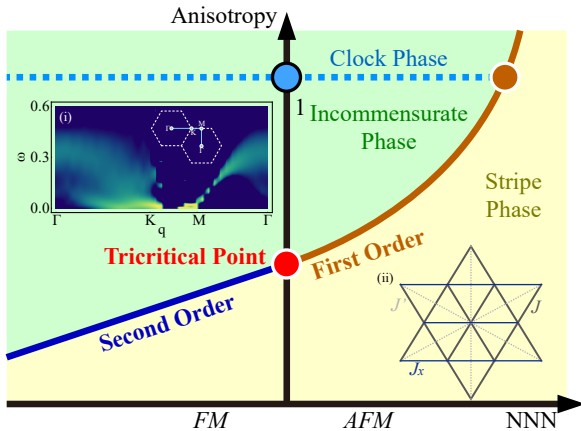

FIG. 1. The schematic phase diagram of the extended transverse-field Ising model in next-nearest-neighbor interaction $J'/J$–spatial anisotropy $J_x/J$ plane. Inset (i) : the $S^z$ spectrum of the low energy excitation mode in the incommensurate phase, including the high-symmetry points marked in the Brillouin zone. Inset (ii) : an illustration of different interactions.

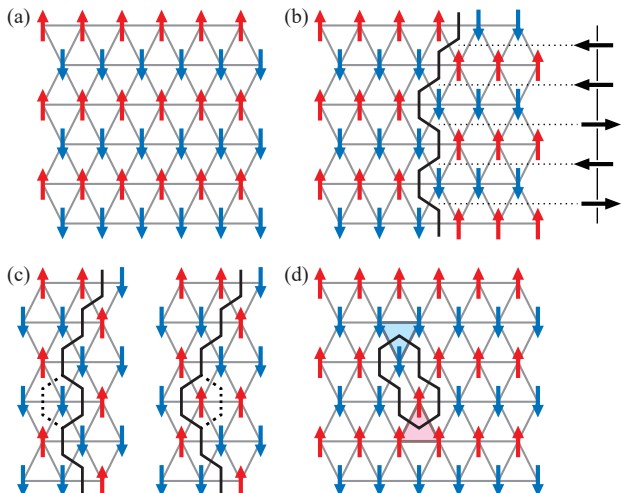

FIG. 2. (a) The spin configuration in stripe phase. (b) The spin configuration with one DW which can be mapped to spin-1/2 chain. (c) The deformation of the DW by flipping spin. (d) The vortex-antivortex pair excitation which break the triangle-rule.

ergy part is mainly contributed from the kink-antikink pair inside the DWs and the collective mode among them. When a next-nearest neighbor (NNN) interaction is introduced, a quantum tricritical point emerges on the incommensurate–stripe transition line. Non-trivially, such tricritical behavior is found due to the changing of effective interactions between the quantum DWs. At last, we enlarged the transverse field and calculated the clock phase's spectrum at a certain parameter region relating to the magnetic material TmMgGaO$_4$ [37–39]. After comparing the branches of different modes, we think that the continuum of the spectrum is due to the merging of vortex mode and DW mode.

This paper is organized as the following. In Section II, we study the anisotropic TFIM without NNN interaction. We establish an effective model describing the DW induced incommensurate phase and provide numerical evidence, including both static and dynamic properties. In Section III, we show that introducing NNN interaction can give rise to a quantum tricriticality. And we also demonstrate such non-trivial critical behavior is caused by an effective long-range inter-DW interaction with two competing terms. In Section IV, we turn to study the isotropic TFIM with NNN interaction, which relates closely to the experiments. From the spectrum, we show the clock phase can also be described by DWs and discuss the origin of the "roton" gap.

## II. INCOMMENSURATE PHASE

The isotropic antiferromagnetic Ising model with transverse field in the triangular lattice is well understood with renormalization group [44, 45], effective field

theory [21, 22] and quantum Monte Carlo simulations [18–20]. In comparison, although the spatial anisotropy widely exists in the real materials, its effect on the frustrated magnetism is still unclear. Therefore, we consider the TFIM with anisotropic nearest neighbor interactions $J_x, J > 0$ of which the Hamiltonian can be presented as

$$H = J_x \sum_{\langle ij \rangle_x} S_i^z S_j^z + J \sum_{\langle ij \rangle_\wedge} S_i^z S_j^z - h \sum_i S_i^x \quad (1)$$

where $h$ is the transverse field, and $\langle ij \rangle_x$ and $\langle ij \rangle_\wedge$ denote the horizontal and diagonal inter-chain bonds, respectively.

### A. Quantum Domain Walls

We consider the Ising limit of $h = 0$. To minimize the ground state energy, similar to the ice-rule in spin ice [46], spins in each triangle need to satisfy the triangle-rule, i.e., one of the spins must be opposite to the other two. When $J_x < J$, all the bonds connecting parallel spins may align in the $x$-directions so that the stripe order is formed, as shown in Fig. 2(a). However, the degeneracy will be recovered at the isotropic point $J_x = J$, because the bonds connecting parallel spins have no preferred directions. The related zero-temperature entropy, called residue entropy [47], is $S/N = 0.323k_B$ [18–20]. We can label all the different bonds of the other triangle-rule satisfying state with short perpendicular bisector if we take the spin configurations in the stripe phase as a reference state. Then, these bisectors can construct the string type topological excitations – domain walls, like

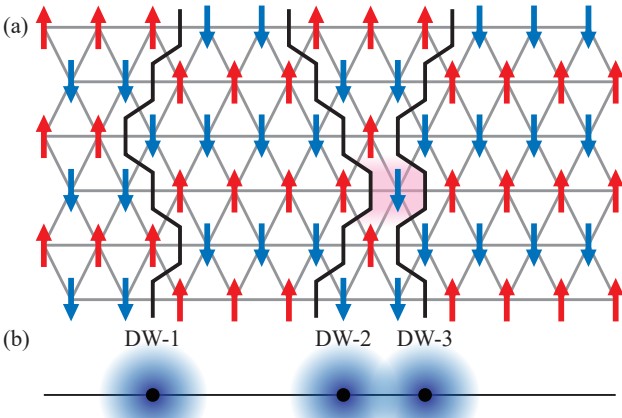

FIG. 3. (a) The spin configuration with three DWs. At low temperature and small transverse field, the DWs can not be crossed with each other (spin inside the red circle can not flip due to triangle-rule), so that there exists effective interactions between DWs. (b) The configuration in upper panel can be taken as three fermions with long-range effective interactions in one dimension.

the single DW case in Fig. 2(b). Meanwhile, the illustration also shows that the DW can be exactly mapped to spin-1/2 chain if we map the left-going (right-going) bisector to effective spin-up (down) [21, 22, 48–50]. The energy gap for single DW excitation is $\Delta = L_y(J-J_x)/2$.

When a finite transverse field $h$ is added, the quantum fluctuation is introduced. As shown in Fig. 2(c), without breaking the triangle-rule, the DW can be deformed by flipping spin. Thus, the transverse field can be taken as the spin exchange interaction between the nearest neighbor site in the effective spin-1/2 chain. On the other hand, from Fig. 2(d), we can find the topological defects in the form of vortex-antivortex pair can also be excited by spin-flippings that break the triangle-rule. Accompanying such excitation, two DWs attached are emerged, but the energy gap proportional to $J$ is so large that the vortex-antivortex pair is hardly excited [21, 22].

In the low-temperature and small transverse field region $T, h \ll J$, the Hilbert space can be approximately restricted to smaller parts in which the triangle-rule is kept so that no vortex is excited. As discussed above, a single DW can be mapped to an effective spin-1/2 XY chain which is exactly solvable with Jordan-Wigner transformation. So the energy of a single DW is

$$E_{\mathrm{DW}} = \Delta + E_{\mathrm{XY}}$$
$$= L_y(J-J_x)/2 - L_y h/\pi. \qquad (2)$$

Then, $E_{\mathrm{DW}}$ can be taken as the effective chemical potential of DW. The DW will be energy preferred compared with the stripe phase when $h/\pi > (J-J_x)/2$. However, the number of DWs will not explosively increase when $E_{\mathrm{DW}} < 0$, because of the effective interactions between them.

Same as the spin-1/2 chain, the size of a single DW's Hilbert space is $2^{L_y}$. However, in the restricted Hilbert space, the DWs can not cross with each other, which is called the non-crossing condition [21, 22], *e.g.*, the DW-2 can not cross with DW-3 in Fig. 3(a). Then, the Hilbert space of each DM will shrink when they get close to each other so that an effective long-range interaction emerges. Therefore, the DWs can be taken as fermions with long-range interaction in one dimension, as shown in Fig. 3(b). If we define the function of interaction $V(r)$, the average energy of DW can be written as

$$\bar{E} = L_y \left( \frac{J - J_x}{2} - \frac{h}{\pi} \right) + V(\bar{r}). \qquad (3)$$

where $\bar{r}$ is the mean distance between DWs. However, the explicit expression of function $V(r)$ is hard to obtain analytically. Furthermore, $E_{\mathrm{XY}}$ only includes the first order perturbation, so the quantitative analysis of the system should turn to the numerical simulations.

## B. Quantum Monte Carlo Simulations

We carry out a quantum Monte Carlo (QMC) calculation using stochastic series expansion (SSE) algorithm [42, 51–53]. We adopt a rectangular periodic boundary condition in both directions. In the simulation, $10^5$ and $2.9 \times 10^6$ Monte Carlo steps are used for equilibrium and measurement. Since updating between different topological sector (changing number of DWs) is difficult in QMC in ultra-low temperature, we equilibrate the configurations into different topological sectors and compare their energies to find the ground state. The unit of energy scale is set to be $J = 1$. To distinguish different quantum phases, we choose two observables – structure factors and DW density.

The structure factor is defined as the Fourier transform of the spin-spin correlator in $z$-direction:

$$S(\mathbf{q}) = \frac{1}{N} \sum_{ij} \langle S_i^z S_j^z \rangle \cos \mathbf{q} \cdot (\mathbf{r}_i - \mathbf{r}_j), \qquad (4)$$

and it is the typical order parameters which reflects the spontaneous breaking of translational symmetry. The DW density is defined as

$$\rho_{\mathrm{DW}} = \frac{1}{N} \sum_{\langle ij \rangle_x} \frac{1 - 4 S_i^z S_j^z}{2} \qquad (5)$$

which effectively counts the number of horizontal bonds connecting opposite spins.

In the weak $x$-coupling limit $J_x \ll J$, as shown in Fig. 4(a), the DW density $\rho_{\mathrm{DW}}$ is approximately zero, indicating the ferromagnetic order forming in the $x$-direction. Meanwhile, the peaks of structure factor $S(\mathbf{q})$ at wave vectors $\mathbf{q} = (\pm 2\pi, 0)$ and $(0, \pm 2\pi/\sqrt{3})$ in inset (i) of Fig. 4(a) confirms the existence of stripe phase. When increasing the coupling $J_x/J$, a jumping point of $\rho_{\mathrm{DW}}$ can

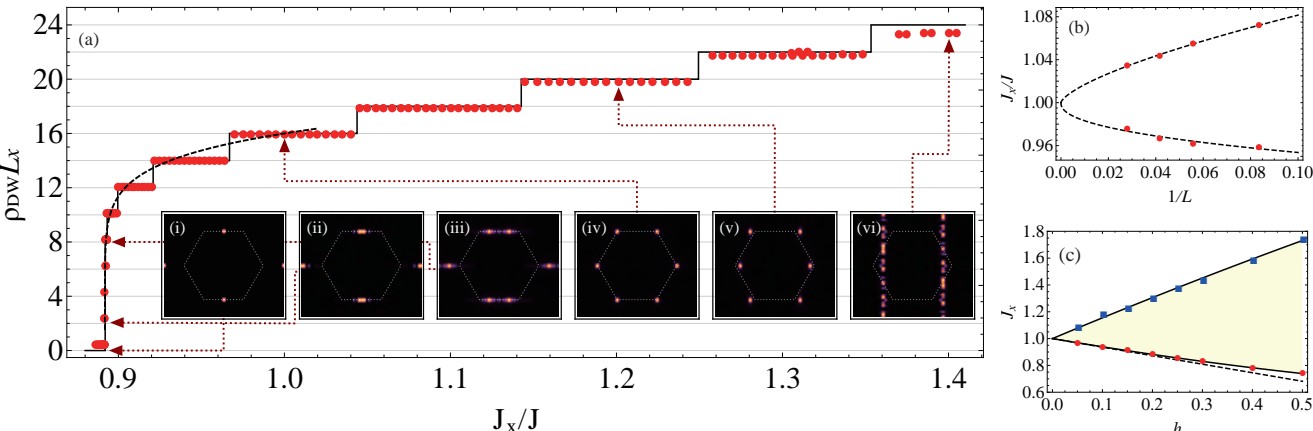

FIG. 4. (a) The relation between DW density $\rho_{\mathrm{DW}}$ and anisotropy $J_x/J$, measured on a $24 \times 24$ lattice at $h = 0.2$ and $\beta = 24$. The dashed line gives the theoretical result in the thermodynamic limit. The grey gridlines show the plateaux of quantization corresponding to even numbers of DWs. Inset: The structural factor at the parameter region with 0, 2, 8, 16, 20, and 24 DWs. The panel (i), (iv), and (vi) corresponds to stripe, clock phase, and decoupled chains, respectively. The dashed white hexagon denotes the first Brillouin zone. (b) The finite-size scaling result for the upper and lower boundary of the $\rho_{\mathrm{DW}} = 2/3$ plateau. (c) The phase diagram on the $h$–$J_x$ plane. The yellowish region denotes the incommensurate phase. The upper and lower critical line separates it with the decoupled chain phase and stripe phase, respectively.

be observed at $J_{x,c} = 0.892$ which is approximately same as the theoretical estimation $J - 2h/\pi = 0.873$ where the single DW energy $E_{\mathrm{DW}}$ becomes zero.

After that, a series of plateaux can be clearly found, which hints strong topological characteristics. Their values $\rho_{\mathrm{DW}}L_x$ are very close to the integer numbers which indicates the number of DWs $N_{\mathrm{DW}}$ in the restricted Hilbert space. Meanwhile, we notice that only even number of DWs exist, and this is due to the periodical boundary conditions we adopted in simulation. Because the TFIM in the triangular lattice can also be mapped to the dimer model [18–20, 54] in the restricted Hilbert space, the states with different number of DWs are corresponding to the topological sectors with different winding numbers.

The broken symmetries of states with multi-DWs can also be checked with the structural factors in the insets of Fig. 4. The clear and sharp peaks indicate that the translational symmetries are spontaneously broken. Meanwhile, when increasing the $\rho_{\mathrm{DW}}$, the centers of peaks move from M points along $x$-direction to K points at $\rho_{\mathrm{DW}} = 2/3$, and continue to move towards $(\pm\pi, 0)$ as $\rho_{\mathrm{DW}} > 2/3$. Such behaviors can be understood when considering the interactions between DWs. If the interaction $V(r)$ is repulsive and decays with distance, the DWs will tend to distribute with equal distance in $x$-direction on average. Then, the emergence of quantum DWs can introduce additional modulation of the domains with wave vectors to be *incommensurate* or called high-order commensurate $\mathbf{q} = (\pm(2 - N_{\mathrm{DW}}/L_x), 0)\pi$ and $(\pm N_{\mathrm{DW}}/L_x, \pm 2/\sqrt{3})\pi$, which are verified by $S(\mathbf{q})$ under high accuracy in the QMC simulations.

Looking closely at the plateaux, we can find their widths increase with $\rho_{\mathrm{DW}}$. It indicates that the repul-

sive interaction $V(r)$ decays faster than linear type. To get the explicit form of the interactions, we consider the total energy

$$E(\rho_{\mathrm{DW}}) = N\rho_{\mathrm{DW}}\left(\frac{J - J_x}{2} - \frac{h}{\pi} + f(\rho_{\mathrm{DW}})\right) \quad (6)$$

where $f(\rho_{\mathrm{DW}}) = V(\bar{r})/L_y$ and $\bar{r} = L_x/N_{\mathrm{DW}} = 1/\rho_{\mathrm{DW}}$. Previous work [34] points out that this interaction should be algebraic $f(\rho_{\mathrm{DW}}) \propto \rho_{\mathrm{DW}}^\alpha$ in the thermodynamic limit. Thus, by extremizing the total energy $\partial E/\partial\rho_{\mathrm{DW}} = 0$, we get the DW density $\rho_{\mathrm{DW}}(J_x) \propto (J_x - J_{x,c})^{1/\alpha}$. Then, with help of Eq. 6, we obtain the exponent $\alpha \approx 7.5(1)$ by fitting the jumping points between different plateaux. Hereafter, because the clock phase at isotropic point $J_x = J$ can also be taken as special commensurate DWs state with $\rho_{\mathrm{DW}} = 2/3$, we can get the explicit form of DW density without fitting the proportional coefficient

$$\rho_{\mathrm{DW}}(J_x) = \frac{2}{3}\left(\frac{J_x - J_{x,c}}{J - J_{x,c}}\right)^{1/\alpha}, \quad (7)$$

which matches well with the finite size numerical results in Fig. 4.

The possible number of plateaux increase while increasing the system size. Meanwhile, their widths shrink. As demonstrated in finite-size scaling (Fig. 4(b)) of the plateau corresponding to the clock phase, the DW density $\rho_{\mathrm{DW}}$ should follow the continuous function as Eq. 7 in the thermodynamic limit. In the meantime, the continuous changes of $\rho_{\mathrm{DW}}$ from 0 to 1 and position of peaks $\mathbf{q}$ from $(\pm 2\pi, 0)$ to $(\pm\pi, 0)$, show a clear mark of continuous phase transition. Besides the clock phase, the state with $\mathbf{q} = (\pm\pi, q_y)$ at $J_x > J$ is also special. The spins

form a spin-density wave in $x$-direction on each horizontal chain, but the inter-chain interactions are so weak that all chains are decoupled in $y$ direction. Thus, the maximum of structural factor locate at $q_x = \pm\pi$ and arbitrary $q_y$ which marks absence of order in $y$-direction (inset (vi) of Fig. 4).

At last, we also take different $h$ values and calculate phase diagram in the $h$–$J_x/J$ plane (Fig. 4c). The region of the incommensurate phase takes the shape of a sector centered at $h = 0, J_x = 1$ point, and expands broader as $h$ increases. For relatively small $h$, the critical lines are linear and agrees with the theoretical value. For larger $h$, the critical line gradually deviates, since higher-order perturbation is not negligible.

### C. Spectroscopy

In the frustrated magnets, the different energy scales are usually not only reflected term of the Hamiltonian, but some of them are also related to the emergent excitations [1, 2]. To figure out different "hidden" modes, we try to study the spectrum of incommensurate phases. We consider the imaginary time correlation of the $z$-component of the spins

$$G^{zz}(\tau, \mathbf{q}) = \frac{1}{\sqrt{N}} \left\langle \sum_{ij} S_i^z(0) S_j^z(\tau) \cos \mathbf{q} \cdot (\mathbf{r}_i - \mathbf{r}_j) \right\rangle \quad (8)$$

in QMC and carried out a stochastic analytical continuation (SAC) to obtain its spectral function $S^{zz}(\omega, \mathbf{q})$. Simulations are implemented on $24 \times 24$ lattice, $\beta = 96$ and $7 \times 10^9$ MCS are utilized in measurement. Two distinct modes can be clearly distinguished in the spectrum (Fig. 5(a)). The higher-lying one is at the energy level of $J$ and corresponds to the vortex mode that violates the triangle-rule. The lower-lying one at the energy level of $h$ shows a clear continuous spectrum and represents the exotic excitation mode corresponding to the DWs. To look more carefully into this mode, we scrutinize the spectrum in the direction along and perpendicular to the DW, $i.e.$, along with $\Gamma M$ and $\Gamma KM$ line, respectively.

The spectrum along the $\Gamma M$ corresponds to the excitations inside the DW, $i.e.$, deformation of the DWs in the form of creation and annihilation of kinks. As discussed in Sec. II A, a DW can be effectively described by an spin-1/2 XY chain, so the continuous spectrum results from the effective Jordan-Wigner fermion excitations. In the dilute DWs case, as shown in Fig. 5(b), the spectral function quantitatively agrees well with that of XY chain (white dash lines), enveloped by two dome-like curves, widest at $\Gamma$ point and shrinking when moving towards at M point. The peak of the lower curve is close to $h = 0.2$, and the maximum of a higher one also equals approximately the theoretical value of $2h$. However, when increasing the density of DWs, we can find the excitation becomes gapped in Fig. 5(d), and we think that it may results from the repulsive interactions between DWs.

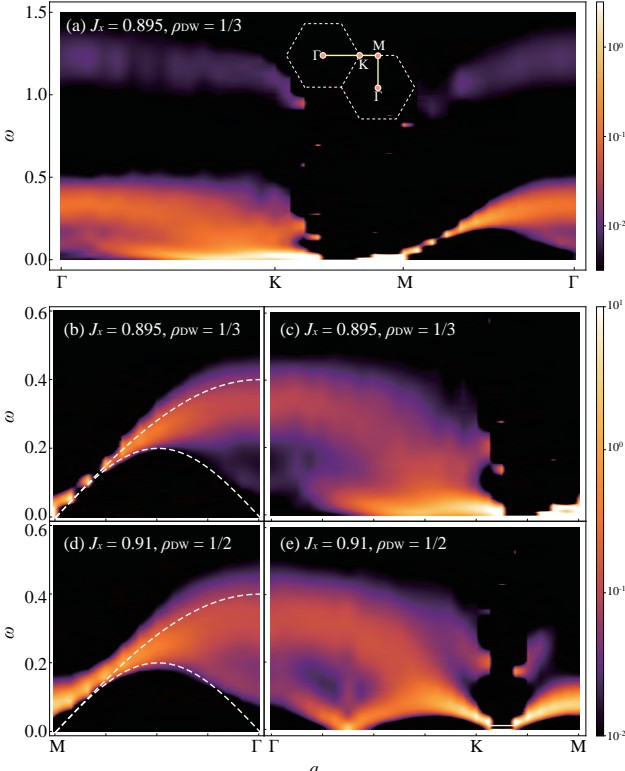

FIG. 5. (a) The spectrum of incommensurate phase measured on a $24 \times 24$ lattice at $J_x = 0.895$, $\rho_{\rm DW} = 1/3$, $h = 0.2$ and $\beta = 96$. The path we take in the Brillouin zone is shown in the inset. (b,c) The zoom-in of low energy region (b) parallel to the DWs along $\Gamma M$ line and (c) perpendicular to the DWs along $\Gamma KM$ line respectively. The low energy excitation spectrum measured at $J_x = 0.91$ and $\rho_{\rm DW} = 1/2$ (d) parallel to the DWs along $\Gamma M$ line and (e) perpendicular to the DWs along $\Gamma KM$ line respectively.

The spectrum along the $\Gamma KM$ line corresponds to the dynamics of DWs in $x$-direction. Firstly, because of the interactions between DWs, the domains can construct the incommensurate spin-wave order, so that the gapped point locates between K and M points at $\omega = 0$ in Fig. 5(c)(e). When turning to the region between this gapped point and $\Gamma$ point, we observe the spectrum also exhibit the continuous features. Especially, the lower envelop line is sensitive to the density of DWs. We can see the zero point moves, and also the peaks become higher while increasing $\rho_{\rm DW}$. Considering that the multi-DWs can be described with spinless fermions with long-range inter-DWs interactions (Fig. 3(b)), these features are reminiscent of the Luttinger liquid.

### III. TRICRITICALITY

In the frustrated magnetic material [37, 38] and trapped ions [41], the influence of long-range interactions can not be neglected. In the Ising limit, such interaction

can also break the degeneracy and result in the stripe phase. However, different from continuous transition through the incommensurate phase in the anisotropic case, the recent QMC simulation [39] points out the direct first-order phase transition from the stripe phase to clock phase. Thus, an intrinsic quantum tricriticality is expected and related to the interplay between long-range interaction and spatial anisotropy. To figure it out, we introduce a next-nearest-neighbor (NNN) interaction with magnitude $J'$ into the Hamiltonian:

$$H = J_x \sum_{\langle ij \rangle_x} S_i^z S_j^z + J \sum_{\langle ij \rangle_\wedge} S_i^z S_j^z + J' \sum_{\langle\langle ij \rangle\rangle} S_i^z S_j^z - h \sum_i S_i^x$$
(9)

where $\langle\langle ij \rangle\rangle$ denotes all the NNN bonds.

By studying the relation between DW density and anisotropy for antiferromagnetic and ferromagnetic NNN interaction, we can check the related change of quantum phase transition order. At the presence of a ferromagnetic NNN interaction shown in Fig. 6(a), the same as no NNN interaction in Fig. 4(a), all the plateaux up to isotropic case are detected. In contract to such clear evidence for continuous phase transition, we can find a notable jump for antiferromagnetic NNN interactions. To be specific, for $L = \beta = 24$ and $J' = h/10 = 0.02$ (red dots in Fig 6(a)), as $J_x$ decrease from 1, the number of DWs $N_{DW}$ changes from 16 to 12, then directly jumps to 0 and skips all the plateaux from $N_{DW} = 2$ to 10. Thus, in the thermodynamic limit, we expect the existence of a first-order transition point $J_{x,c}$ where DW density jump from a critical value $\rho_c$ to zero for antiferromagnetic NNN interactions.

The different types of incommensurate–stripe phase transition indicates that a tricritical point should locate around $J'/J \approx 0$. Usually, the tricritical point is related to the order parameter field with multi-components, such as coplanar phase [55, 56], or binary Bose mixtures [57]. However, here the tricriticality has its topological feature. It divides the parameter region where topological sectors of DWs are complete or not. Such a feature is similar to the incommensurate phase in high-$T_c$ superconductivity, where the first-order phase transition is probably due to the NNN interactions between electrons [27].

To lock the tricritical point, we calculated the phase diagram in $J_x$–$J'$ plane shown in Fig. 6(b). For finite size systems, the incommensurate phase is divided into regions with quantized numbers of DWs, separated by jumping lines where the energy of $N_{DW} = k$ and $k + 2$ is equal. The transition line is determined in such a way that the energy of the stripe phase equals the lowest energy of the incommensurate phase among all the quantized plateaux. For negative $J'$, all the jumping lines are well determined. While $J'$ increases to be positive, the jumping lines gradually cross the transition line. Accompanying crossing, the plateaux will disappear one after another. Thus, the first plateau $N_{DW} = 2$ will disappear at finite size tricritical point which can be obtain by solve the equation $E(\rho_{DW} = 0; J', J_x) = E(\rho_{DW} =$

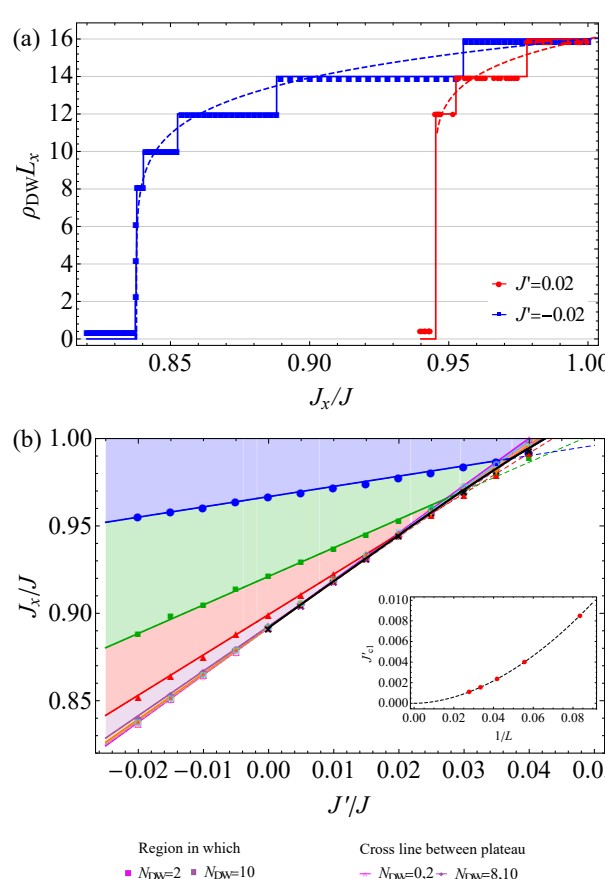

FIG. 6. (a) DW density $\rho_{DW}$ versus anisotropy $J_x/J$ relation with NNN interaction at $L = \beta = 24$ and $h/J = 0.2$. The dashed line gives the theoretical results in thermodynamic limit. (b) The numerical finite size phase diagram in NNN interaction–anisotropy plane measured at $L = \beta = 24$ and $h/J = 0.2$. The filled regions shows parameter regions with different number of DWs and the colored lines shows the jumping lines between them. The black line shows the first order transition line. Inset: The expected behavior of the disappearing point of the first plateau at finite size towards thermodynamic limit given by $E(\rho_{DW} = 0) = E(\rho_{DW} = 2/L) = E(\rho_{DW} = 4/L)$.

$2/L_x; J', J_x) = E(\rho_{DW} = 4/L_x; J', J_x)$. Moving towards the thermodynamic limit, this point is found to gradually approach $J' = 0$ in Fig. 6(b). Then, the question turns to how to understand such tricriticality through the effective DWs paradigm.

Taken into account the energy intra- and inter-DWs, our effective model yields that the energy of DWs can be written as $E(\rho_{DW}) \propto \rho_{DW}(A + B\rho_{DW}^\alpha)$, which is always concave and only have one extremum. However, to create two competing phases apart requires at least two extrema, so the ansatz of interactions between DWs needs to be adjusted at the presence of the NNN interaction. In Fig. 6(a), the insertion of single DW produces

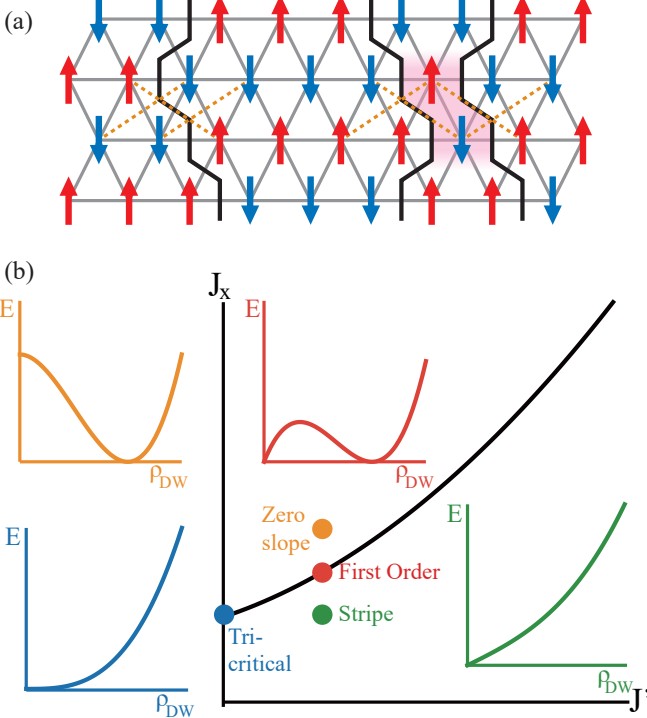

(a)

(b)

FIG. 7. (a) The spin configuration with three DWs. The orange lines show the intra- and inter-DW energy induced by NNN interactions. (b) An illustration of the energy versus DW density curve in vicinity of the tricritical point.

energy cost $3J'/2$ per DW length, while two adjacent DWs cost energy $2J'$ per DW length, which is less than two individual DWs. Therefore, antiferromagnetic NNN interaction produces an effective attractive long-range interaction between DWs, while ferromagnetic interaction produces a repulsive one. Thus, we suggest a straightforward term $-C(J')\rho_{\text{DW}}^{\gamma}$ be added into the DW energy

$$E(\rho_{\text{DW}}) = N\rho_{\text{DW}}(A + B(J')\rho_{\text{DW}}^{\alpha} - C(J')\rho_{\text{DW}}^{\gamma}). \quad (10)$$

Because the energy above should return to no NNN interaction case when setting $J' = 0$, we can get $C(0) = 0$ and $B(0) \neq 0$. Keeping the leading order, Eq. 10 should be rewritten as

$$E(\rho_{\text{DW}}) \approx N\rho_{\text{DW}}(A + B_0\rho_{\text{DW}}^{\alpha} - C_1 J'\rho_{\text{DW}}^{\gamma}). \quad (11)$$

If $C_1$ is positive and $\alpha > \gamma$, the appearance of tricriticality can be well explained, as demonstrated in Fig. 7(b). When $J'$ is small and positive, the two competing interactions result in an inflection point in the $E$–$\rho_{\text{DW}}$ curve. At small $J_x$, the stripe phase has the lowest energy. Then, when increasing $J_x$, due to the last term of $E(\rho_{\text{DW}})$, another energy minimum appears for the incommensurate phase with certain non-zero DW density. The energy gap to the stripe phase will be closed at transition point $J_{x,c}$. The jumping of DW density $\rho_{\text{DW}}$ indicates phase transition is the first order. When $J' < 0$,

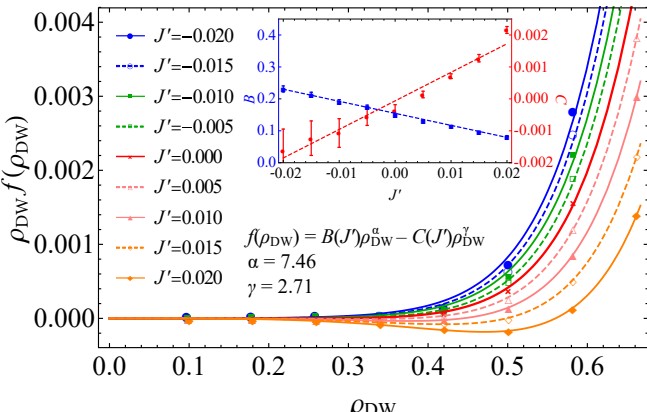

FIG. 8. The numerical result of DW interaction versus DW density relation for different NNN interaction $J'$. The dashed and solid lines show the result from the fitting. Inset: The interaction coefficient $B$ and $C$ versus NNN interaction $J'$.

the last term takes the same effect as the second term, and the incommensurate–stripe phase transition remains second order.

In order to numerically determine the interaction coefficients in $f(\rho_{\text{DW}}) = B(J')\rho_{\text{DW}}^{\alpha} - C(J')\rho_{\text{DW}}^{\gamma}$, we calculated the energy $E(\rho_{\text{DW}})$ versus DW density for different $J'$. In finite size system, the linear term can be obtained by finding $E_l = E(\rho_{\text{DW}} = 0) = E(\rho_{\text{DW}} = 2/L_x)$ (which equals the critical point when $J' < 0$). Then, after substracting the linear term from $E(\rho_{\text{DW}})$, we can get the finite size value of $\rho_{\text{DW}}f(\rho_{\text{DW}})$. As shown in Fig. 8, $\rho_{\text{DW}}f(\rho_{\text{DW}})$ fits well with the exponents $\alpha = 7.5(1)$ and $\gamma = 2.7(3)$, and $\alpha$ larger than $\gamma$ coincides with our prediction. Meanwhile, from the inset of Fig. 8, we can find the leading order approximation of the coefficients $B$ and $C$ are good enough. Thus, the numerical results strongly support our theoretical analysis: *the tricriticality of the incommensurate phase is caused by the competing of effective long-range inter-DW interactions with different power exponents.*

## IV.   CLOCK PHASE

Recently there have been some experimental and numerical studies on the isotropic Ising model with NNN interaction [37–39]. Without the external magnetic field, the ground state is found to be clock phase. Meanwhile, the numerical simulation discovers an excitation mode, which is identified as "roton". Based on the effective field theory [21, 22] and the renormalization group [44, 45], the thermal phase transition from the clock phase to the Kosterlitz-Thouless phase should be Berezinskii-Kosterlitz-Thouless (BKT) type. On the other hand, the Kosterlitz-Thouless phase will undergo a second BKT phase transition to the paramagnetic phase by increasing the temperature, and the excitation mode is vortex-antivortex pair which breaks the triangle-rule.

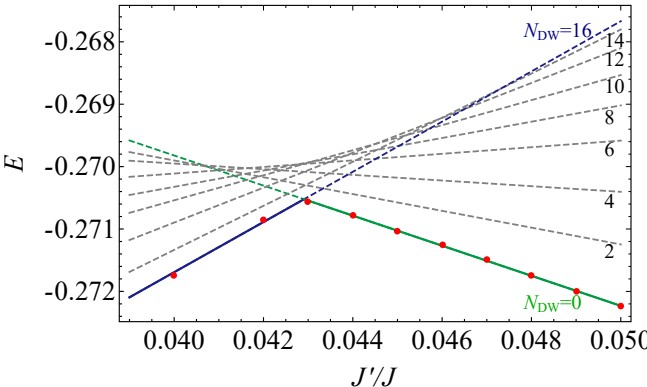

FIG. 9. The energy as a function of the NNN interaction $J'/J$ at $h/J = 0.2$. The black solid line and the red dots denote the ground state energy, and the higher-lying gray dashed lines denote the energy of states with different numbers of DWs.

To detect the different modes of clock phase, at last, we turn to study the isotropic case $J_x = J$ of Hamiltonian Eq. 9.

For small $h/J$, the effective theory of DW can still work well [21, 22]. In Fig. 9, the energy–NNN interaction relation at $h/J = 0.2$ shows a sharp wedge between stripe and clock phase. All the states with DW density other than 0 and 2/3 are higher-lying and excited states. Along the isotropic line $J_x/J = 1$ (dash line in Fig. 1), the DW induced incommensurate phase is locked to a commensurate point and takes the form of clock phase. The DW density $\rho_{DW} = 2/3$ doesn't change until crossing the first-order transition point and entering the stripe phase.

The characteristics of the spectrum (Fig. 10(a)) of the incommensurate phase still pertain in the clock phase. A low-lying DW mode is distinct. Along the $\Gamma M$ line, the excitation takes the form of effective kink-antikink, and its continuous spectrum conforms well with that of spin-1/2 XY chain. Along the $\Gamma KM$ line, the excitation corresponds to the interaction between DWs. The average DW interval is $l_x = 3/2$, so that the static peak of the structure factor locates at $q_x = 2\pi/l_x = 4\pi/3$, which agrees well with the numerical results.

However, this mode has never been observed either in experimental or theoretical works. This is because effective transverse field $h$ in actual materials is much higher than the region in which we can observe the DW mode. Thus, we increase the transverse field from $h = 0.2$ to $h = 0.5$, which is close to the experimental values shown in Fig. 10. The low-lying DW mode split into two branches at $h \geq 0.2$, and the weight gradually transfers from the lower branch to the higher branch at $h \geq 0.35$. This is because when $h$ becomes larger, the higher-order perturbations in constructing our effective model become prominent, and they may deform the spectrum of effective DW Hamiltonian. Meanwhile, the energy of DW mode becomes higher as $h$ increases, and the DW mode gradually merges with the higher-lying mode when $h$ and

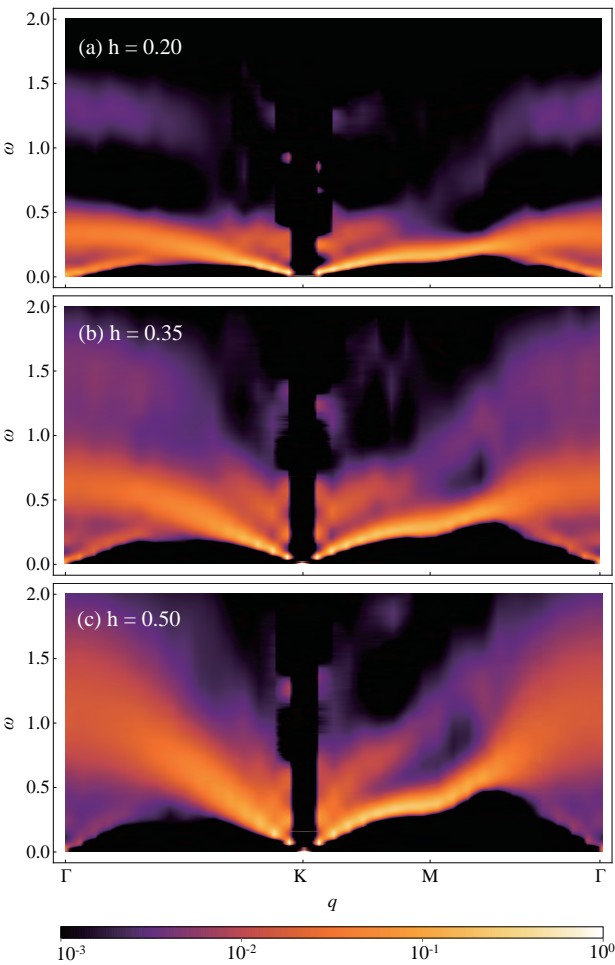

FIG. 10. The spectrum of isotropic clock phase with different transverse field strength, measured on a $24 \times 24$ lattice at (a) $h = 0.2$, (b) $h = 0.35$ and (c) $h = 0.5$ respectively and $J' = 0.1h$, $\beta = 24$.

$J$ reaches the same magnitude.

The higher-lying mode corresponds to the excitations that break the triangle-rule. In the representation of DWs, such excitation corresponds to confined ends of the DWs. Those confined ends cost great energy when $h \ll J'$. As $h$ increases into the same magnitude of $J$, the energy costs of creating and annihilating a pair of DWs is no longer much higher than the cost of deforming a DW. Hence, these two modes couple together, resulting in the merge shown in the spectrum.

## V. CONCLUSION AND OUTLOOK

To summarize, we study the phase diagram of the extended anisotropic Ising model in the triangular lattice with the transverse field. We find the quantum phase with incommensurate order when considering the spatial anisotropy, which can be explained in terms of quantum

topological defect – domain wall excitations. The DW density and structure factor indirectly support the existence of DW. On the other hand, the low-lying modes in the incommensurate phase's spectrum directly demonstrate that: (i) the dynamics inside DW can be described with spin-1/2 XY model and (ii) the multi-DWs can be taken as many spinless fermions with long-range interactions. The phase transition between the incommensurate phase and the stripe phase is found to be continuous. Then, we check the influence of the NNN interaction and find the phase transition changes into first order in the antiferromagnetic side. The related quantum tricritical point is approximately at no NNN interaction point $J' = 0$. To figure out its origin, we propose a new ansatz of the effective inter-DW interaction with form $\frac{B}{r^\alpha} - \frac{C}{r^\gamma}$. After fitting with the numerical results, we obtain the power exponents are $\alpha = 7.5(1)$ and $\gamma = 2.7(3)$, and the leading order approximation is good enough for explaining such exotic quantum tricriticality. Last, we revisit the isotropic case with NNN interaction, which is the most experimentally relevant region. Our simulation also supports the first order clock–stripe phase transitions the same as recent numerical conclusion [39]. In comparison, basing on our analysis of different modes in the strongly correlated region $h \ll J$, we regard the so-called "roton" mode as contributed by both low-lying DW mode and high-lying vortex mode which violates the triangle rule.

The incommensurability plays a critical role in not only frustrated magnetism but also the high-$T_c$ superconductors. Our work provides insightful predictions of the real system. For frustrated magnetism, the incommensurate phase should be found when introducing the spatial anisotropy, such as adding pressure. In the case of TmMgGaO$_4$ [38] [58], we expect that by replacing Tm in the TmMgGaO$_4$ material with other rare earth metals, one can obtain an Ising magnet with smaller energy splitting $h$ so that the continuous DW mode spectrum can be detected more clearly. The tricritical point may be verified by using the trapped ions. The NNN interaction can be detuned by changing the wavelength of the optical lattice, and the spatial anisotropy can be realized by making the wavelength of lasers different in different directions. Besides, the first-order phase transition resulting from the NNN interaction may shed light on a similar effect on the high-$T_c$ cuprate superconductors.

## ACKNOWLEDGEMENTS

We wish to thank Yang Qi, Jun Zhao and Zi-Yang Meng for the fruitful discussions. This work was supported by the National Key Research and Development Program of China (Grants No. 2017YFA0304204 and No. 2016YFA0300504), the National Natural Science Foundation of China (Grants No. 11625416 and No. 11474064), and the Shanghai Municipal Government (Grants No. 19XD1400700 and No. 19JC1412702). X.-F. Z. acknowledges funding from the National Science Foundation of China under Grants No. 11804034, No. 11874094 and No. 11947406, and Grant No. cstc2018jcyjAX0399 by Chongqing Natural Science Foundation. Z.Z. acknowledges support from the CURE (H.-C. Chin and T.-D. Lee Chinese Undergraduate Research Endowment) (19925) and National University Student Innovation Program (19925).

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
