# Peer review of "Quantum tricriticality of incommensurate phase induced by quantum domain walls in frustrated Ising magnetism"

_SciPost Physics_

## Round 2 · Referee Report · Anonymous (Referee 1) · 2022-9-2

Report

In the paper, Zhou et al. studied the ground state of the triangular-lattice quantum Ising model with spatial anisotropy and NNN interactions $J’$, focusing on the quantum tricriticality induced by effective interactions between quantum strings. The quantum Ising model, or transverse-field Ising model, has been extensively studied for many years, but recently it attracts lots of attention in relation to the development of artificial quantum simulators with trapped ions, Rydberg atom arrays, etc. Based on the numerical QMC calculations, the authors investigated here the $J’ $ dependence of the energy of strings, assuming that it is given in the form with two different algebraic-decaying terms. As a results, it is concluded that “the tricriticality of the incommensurate phase is caused by the competing of effective long-range inter-string interactions with different power exponents.”

The paper is scientifically sounds and well written. However, I would recommend the manuscript for publication only after the points listed below are properly addressed.

--- page 4: In the first sentence, the authors state that all the bonds connecting parallel spins are aligned along the x-direction, forming stripe phase, for $J_x < J$. However, the actual ground state can be incommensurate phase when $J_x/J$ is larger than a certain critical value between 0 and 1. This is confusing.

--- page 10: The authors state that the leading order approximation of $B$ and $C$ (namely, constant and linear term, respectively) is good enough. However, according to the inset of Fig. 6(c), the liner term should also be considered for $B$, like $B \sim B_0 + B_1 J’$.

--- Although I understood that the energy can be well approximated in the form of Eq. (10) (except for the above-mentioned point), I could not get the idea of why it yielded the conclusion that the effective inter-string interaction can be written in the form $B/r^\alpha – C/r^\gamma$ (what is $r$?).

--- In the sketch of the phase diagram in Fig. 1, the tricritical point is located just on the vertical axis. However, it is not the case, is it? This is somewhat misleading. Also, using the effective energy functional Eq. (10) with numerically fitted values of coefficients, I think that the authors can estimate the location of the tricritical point ($J_x$ and $J’$) in a more quantitative way. It should help to improve the quality and completeness of the paper.
  • validity: -
  • significance: -
  • originality: -
  • clarity: -
  • formatting: -
  • grammar: -

Author:  Zheng Zhou  on 2022-10-05  [id 2878]

(in reply to Report 1 on 2022-09-02)

We thank the referee for his positive evaluation as well as the critical comments that have helped us improve the manuscript. In the following we give a point-by-point reply to the comments. A more properly formatted version of the reply in LaTeX can be found attached in the PDF file.

  1. In the second paragraph of Sec. 2, we start by considering the classical Ising limit h=0, i.e., the vertical axis of Fig. 4(c). Whereas the incommensurate phase only emerges at finite transverse field, at h=0 all the J_x<J region belongs to the stripe phase by optimising the energy classically. We start considering the quantum case only from the third paragraph, which is after the position the referee mentions. In the revised manuscript, we make clarification by replacing 'in the anisotropic case J_x<J' by 'in the classical anisotropic case J_x<J and h=0'

  2. The referee is correct that for an accurate description, it is better to include a linear term in B. However, approximating B to a constant suffices to reproduce the qualitative result. On one hand, unlike C which changes sign with J', the J'-dependence of B is not other drastic. On the other hand, the B-term interaction originates from the hinder of motion when strings are nearby; this mechanism does not depend on the presence of J'; by contrast, the C-term interaction directly comes from the NNN coupling. By approximating B=B_0 and C=C_1J', one can find that the stripe-incommensurate phase transition is first-order when J'>0 and continuous when J'<0, and there is therefore a multicritical point at J'=0.

To clarify, in the revised manuscript, we add some related discussion: 'We also note that for a more accurate description, B(J') should be approximated as B(J')\approx B_0+B_1J' to take into account the fact that the NNN coupling modifies the vibration of string, but this modification does not change the scenario qualitatively and can be therefore considered as a higher-order term.'

  1. Firstly, r is the distance between two nearby strings. To calculate the average energy of quantum string, we replace r by its average value \bar{r}=L\rho_{QS}.

Regarding the interaction energy ansatz V(r)=B/r^\alpha-C/r^\gamma, we here give a more detailed explanation. There are two mechanisms of string interaction: the first, denoted V_h(r), is a repulsion from the hinder of motion when strings are nearby; the second, denoted V_{J'}(r), comes from the fact that the insertion of single string produces energy cost 3J'/2 per string length, while two adjacent strings cost energy 2J' per string length, which is different than two individual strings; therefore we can write V(r)=V_h(r)+V_{J'}(r). Originally, these two mechanisms all act in short-range; the vibration of strings then turns these short-range interactions into long-range ones. The exact determination of the form of V_{J'}(r) and V_h(r) is a difficult question. In a former work [3], the repulsion V_h(r) from motion hinder has been determined to follow a power-law behaviour in the incommensurate phase a similar model (hardcore Bose-Hubbard model) by fitting the jumping points of the plateaux. As an extension, it is most natural and simplest to assume that V_{J'} also follow a power law. We therefore write down the ansatz V(r)=B/r^\alpha-C/r^\gamma. As an evidence, this ansatz fits well with our numerical result (Fig. 6(c)), as explained in the main text.

To clarify, we have also added the above paragraph to the main text, added the sentence 'where r is the distance between two nearby strings' to the Conclusion section, and deleted the exact expression from the abstract to avoid confusion.

The tricritical point in fact locates on the vertical axis J'=0 exactly. The numerical evidence is provided in Fig. 5(b) inset. We determine the tricritical point at finite size as the J' value where the first plateau disappears, i.e., E(\rho_{QS}=0)=E(\rho_{QS}=2/L)=E(\rho_{QS}=4/L), and performed a finite size scaling. The result shows that the tricritical point should lie on the vertical axis J'_c=0.

Intuitively, the tricriticality between first order and continuous transition line is driven by the string interaction mediated by J' (C-term) changing from attractive to repulsive. As C-term is mediated by the NNN coupling J', it is natural that the tricritical point locates at the zero-point of this coupling.

In the manuscript, it is also mentioned that 'Moving towards the thermodynamic limit, this point is found to gradually approach J'=0 in Fig. 5(b) inset'

Attachment:

Reply_1.pdf

---

## Round 2 · Referee Report · Anonymous (Referee 2) · 2022-9-9

Report

The authors have studied the ground state phases of the transverse field Ising model with nearest and next nearest neighbor interactions on an anisotropic triangular model at weak transverse fields. They develop a description of the ground state order in terms of strings separating domains of columnar AFM order. Each string behaves as a S=1/2 XY chain with long-range interaction. The central result of the work is the detection of an incommensurate phase and the appearance of a tricritical point when NNN interactions are introduced.

The problem is interesting and the results are intriguing. However, in the opinion of the present referee, the manuscript can be improved before publication. In particular, can the authors address the following?

  1. How is the sign problem avoided in SSE? Previous QMC studies (13, 33, https://www.nature.com/articles/s41467-020-14907-8) all use special trick with path integral / world line QMC to avoid the sign problem where the coupling in the imaginary time direction is shown to be ferromagnetic. Sandvik's work (PhysRevE 68 056701 (2003) using SSE on TFIM with long range interactions use a bipartite lattice. Since the authors are probably the first to use SSE for TFIM on a triangular lattice, a discussion on the method would greatly add to the manuscript.
  2. In the phase diagram 4(c), can the authors discuss the stripe phase? It appears to be deep inside the incommensurate phase.
  3. How do the authors conclude that V(r) decays faster than linear? Doesn't the hard core constraint generalize to fast decaying interaction in the presence of quantum fluctuation (h > 0)?
  4. Does "vibration" of the string refer to the process defined in Fig. 2(c)?
  5. It is not clear how the authors arrive at the functional dependence of the energy of the strings.
  6. Should the width of the plateaus reduce to zero in the thermodynamic limit? What about the stripe phase? Does it have any finite-J_x extent in the thermodynamic limit?

To conclude, the authors have introduced an interesting problem. If they can address the above issues satisfatorily, the manuscript can be considered for publication.

  • validity: -
  • significance: -
  • originality: -
  • clarity: -
  • formatting: -
  • grammar: -

Author:  Zheng Zhou  on 2022-10-05  [id 2880]

(in reply to Report 2 on 2022-09-09)
Category:
remark
answer to question

We thank the referee for his positive evaluation as well as the critical comments that have helped us improve the manuscript. In the following we give a point-by-point reply to the comments. A more properly formatted version of the reply in LateX can be found attached in the PDF file.

1 . The transverse field Ising model does not have sign problem regardless of the lattice, as one can always add a constant to eliminate the negative matrix elements. E.g., the AFM TFIM

H=J\sum_{<ij>}S_i^zS_j^z-\Gamma\sum_iS_i^x

can be rewritten in the following by adding a constant energy shift

H=J\sum_{<ij>}(S_i^zS_j^z-1/4)-\Gamma\sum_i(S_i^x+1/2)\\
=-\sum_{<ij>}H^d_{ij}-\sum_iH^o_i,

where the local operators and their matrix elements are

H^d_{ij}=J(-S_i^zS_j^z+1/4)
<↑↓|H^d_{ij}|↑↓>=<↓↑|H^d_{ij}|↓↑>=J/2,(others)=0
H^o_i=\Gamma(S_i^x+1/2)
<↑|H^o_i|↑>=<↑|H^o_i|↓>=<↓|H^o_i|↑>=<↓|H^o_i|↓>=\Gamma/2

All the matrix elements are non-negative; there is therefore no sign problem. In addition, some reference [Phys. Rev. B 98, 174433 (2018)] also applies SSE to non-bipartite lattices; the original reference (Ref. [48]) does not restrict the simulation to bipartite lattice.

2 . The stripe phase is actually not incommensurate, as its Bragg peak situates at the commensurate M point in the momentum space (Fig. 3(a)i). The real-space spin configuration in the stripe phase has parallel spins in the same row and alternating spin-ups and spin-downs in adjacent rows (Fig. 2(a)). Unlike the incommensurate phase which only appears in the presence of quantum term h, the stripe phase optimises the energy in the classical limit h=0 at J_x<J. The reason we choose the stripe state as the reference state is that any local operation acting upon this state violates the local constraint, hence there are no low-energy excitations within the stripe bulk.

We have also added a brief discussion 'The ordering momentum is commensurate. The spin alignment alternates each row in the real space configuration [Fig. 2(a)]. This phase extends to J_x\to-\infty and persists at the classical limit h=0.' to Section 3, Paragraph 2.

3 . We provide a detailed description on how we determine the form of the interaction between strings in the Reply 5. The power of the interaction is determined by fitting. In the presence of quantum fluctuation, the hardcore constraint generates a power-law dynamic interaction.

4 . Yes. In the revised manuscript, we have added the reference to Fig. 2(c) when mentioning the 'vibration'.

5 . As explained in Sec. 2, the energy of the string consists two parts. The first is the energy of single quantum string (Eq. (2)), which consists of the energy gap \Delta and the kinetic energy obtained from mapping the string configuration to an effective XY chain; the second is the interaction between adjacent strings V(\bar{r}), where V(r) is the interaction energy between adjacent strings, r is the distance between two nearby strings, and \bar{r}=L\rho_{QS} is its average value.

Regarding the interaction energy ansatz V(r)=B/r^\alpha-C/r^\gamma, we here give a more detailed explanation. There are two mechanisms of string interaction: the first, denoted V_h(r), is a repulsion from the hinder of motion when strings are nearby; the second, denoted V_{J'}(r), comes from the fact that the insertion of single string produces energy cost 3J'/2 per string length, while two adjacent strings cost energy 2J' per string length, which is different than two individual strings; therefore we can write V(r)=V_h(r)+V_{J'}(r). Originally, these two mechanisms all act in short-range; the vibration of strings then turns these short-range interactions into long-range ones. The exact determination of the form of V_{J'}(r) and V_h(r) is a difficult question. In a former work [3], the repulsion V_h(r) from motion hinder has been determined to follow a power-law behaviour in the incommensurate phase a similar model (hardcore Bose-Hubbard model) by fitting the jumping points of the plateaux. As an extension, it is most natural and simplest to assume that V_{J'} also follow a power law. We therefore write down the ansatz V(r)=B/r^\alpha-C/r^\gamma. As an evidence, this ansatz fits well with our numerical result (Fig. 6(c)), as explained in the main text.

To clarify, we have also added the above paragraph to the main text, added the sentence 'where r is the distance between two nearby strings' to the Conclusion section, and deleted the exact expression from the abstract to avoid confusion.

6 . The width of the plateaux scales to zero in the thermodynamic limit. As an evidence, in the manuscript, we did the finite size scaling for a specific plateau (\rho_{QS}=2/3) and find that the ends of the plateau extrapolate to the same point [Fig. 4(b)]. The stripe phase will remain a finite region in the thermodynamic limit. As an intuitive argument, when the anisotropy is much stronger than the transverse field J-J_x\gg h, the stripe phase is always favoured. The estimation of the stripe-incommensurate critical point J_{x,c}=J-2h/\pi+{O}(h^2) applies also to the thermodynamic limit.

To clarify, we have also added a sentence 'As demonstrated in finite-size scaling (Fig. 4(b)) of the plateau corresponding to the clock phase, the width of each plateaux scales to zero the string density \rho_{QS} should follow the continuous function (7) in the thermodynamic limit.' to the related discussion.

We hope that with these changes made, the Referee could agree that the manuscript is suitable for further consideration for SciPost Phys.

Attachment:

Reply_2.pdf

---

## Round 3 · Referee Report · Anonymous (Referee 1) · 2022-10-6

Report

The authors have properly addressed the comments and questions I raised in the previous report. Now, I would recommend the paper for publication in the present form.

---

## Round 3 · Referee Report · Anonymous (Referee 3) · 2022-10-14

Strengths

1.) Interesting relevant model for frustration and criticality

2.) Groundbreaking analytic analysis in terms of interacting strings

3.) Convincing large scale QMC simulations

Weaknesses

1.) some discussions are missing or not detailed enough (see report)

Report

The authors consider the transverse field Ising model on a triangular lattice,
which is an interesting model for rich critical behavior due to the interplay
of frustration and quantum effects. By considering an anisotropic coupling
it is possible to explain much of the behavior with an effective description in
terms of interacting strings, which is supported by large scale numerical simulations.

Apart from the necessary changes (below), I find the paper truly convincing. The work
meets the acceptance criteria and should be published after those changes have
been considered

Requested changes

1.) In the description of the construction of strings on page 4, I could not understand
the following sentence: "To avoid creation of triangle-rule-breaking defects (also known as
spinon topological defects), each bisector within the string can only choose left-going
or rightgoing directions" What is meant by "bisector"? Where do I see the directions in
Fig 2? I recommend that the explanation is expanded in more detail.

2.) In Eqs. (2) the energy of the quantum strings are defined. It should be
explained if there is a kinetic energy as well or why it can be neglected.

3.) In Eq. (5) the string density is defined, which appears to be quantized
in the numerical simulations. Is it a conserved quantity or is there another
explanation for this discrete behavior (finite size effect)? The change of the
peak position is argued to become continuous in the thermodynamic limit, but does
(density x length) remain quantized?

4.) The assumption of a power law interaction in Eq. (9) is not rigorously motivated
as previous referees also commented.
A discussion would be useful how important this assumed form is to the final outcome,
or if other forms of two competing interactions have also been tried. The clear evidence
of a long range attractive contribution to the interaction is surprising and interesting.
What could be the mechanism? The newly inserted paragraph does not explain why
one part is attractive.

5.) The relation to the hard-core boson model in Ref. [28] should be discussed
in more detail, which seems to follow similar physics. What is different?
Is the universal critical behavior the same?

6.) Editorial changes: Refs. [3] and [51] are identical.
Please check for spelling mistakes ("incommensurte"on p.2)
and spurious articles (remove "the" in front of QMC simulations).

  • validity: top
  • significance: high
  • originality: top
  • clarity: high
  • formatting: good
  • grammar: good

Author:  Zheng Zhou  on 2022-10-26  [id 2954]

(in reply to Report 2 on 2022-10-14)
Category:
answer to question

We thank the Referee for the recommendation for publication as well as the useful comments and suggestions, which helped us to improve the manuscript. In the following we give a point-by-point reply to these comments. A more properly formatted version of the reply in LateX can be found attached in the PDF file.

1 . We thank the Referee for pointing out our ambiguous expression. In the revised manuscript, we have replaced the word 'bisector' by 'segment', and marked it in the Fig. 2 by the green and purple left and right-pointing arrows.

2 . In fact, the vibration of the segment is described by an XY-chain and the kinetic energy corresponds to the energy of the XY-chain E_{XY} and is included in Eq. (2). In the revised manuscript, we have made that clear by adding a sentence 'where \Delta is the energy gap and E_{XY} is the kinetic energy given by solving the effective spin-1/2 XY-chain'.

3 . The quantisation of string density is due to finite size effect. As the number of strings must be a even integer under periodic boundary condition, the string density must then be 2\mathbb{Z}/L_x. In the limit of small quantum fluctuation where the triangle-rule cannot be violated, any local operation cannot change the number of quantum strings, so the string density is also conserved. In the thermodynamic limit, the quantisation step 2/L_y becomes infinitesimall, so the string density becomes continuous.

4 . For the justification of the ansatz, there has in fact been a long debate between whether the interaction between strings should be exponential or power-law [J. Zaanen, Phys. Rev. B 40, 7391(R) (1989)]. We have tried both ansatz
V_1(r)=B(J')/r^๐›ผ-C(J')/r^๐›พ
V_2(r)=B(J')e^{-r/๐œ‰_1}-C(J')e^{-r/๐œ‰_2}
for the second ansatz, the optimal parameters are calculated to be ๐œ‰_1=0.19 and ๐œ‰_2=0.89. The sum of residual squared of the power-law ansatz is 4.35ร—10^{-7} and for the exponential ansatz is 5.73ร—10^{-7}. We therefore adopted the power-law ansatz in the manuscript. The form of the ansatz does not affect our qualitative result as long as in V(r)=V_h(r)+V_{J'}(r), V_h(r) decays faster than V_{J'}(r), and V_{J'}(r) changes sign when J'=0. We also note that the mechanism of the repulsion V_h(r) is similar to the hard-core boson model, so there is no reason to expect that the form of interaction should be different. We have also added the discussion to a footnote in the main text.

For the reason of the attractive interaction, we added some further explanation to the newly added paragraph. At the presence of J', there is an additional mechanism of string interaction: apart from the repulsion from the hinder of motion when strings are nearby denoted V_h(r), the second, denoted V_{J'}(r), comes from the fact that the insertion of single string produces energy cost 3J'/2 per string length, while two adjacent strings cost energy 2J' per string length, which is different than two individual strings [Fig. 6(a)]. Therefore, when two string segments are adjacent, there is an extra energy gain of J' when J'>0, resulting in an attractive interaction, while when J'<0, this becomes an energy cost of |J'|, resulting in a repulsive interaction.

5 . In the hardcore boson model, the interaction power is calculated to be ๐›ผ=4.0(1), which is different from our result ๐›ผ=7.5(1). The difference is because of the different manners of string vibration. In the hardcore boson model, the vibration of the string is described by an XY-chain with only next-nearest-neighbour interaction, which is different from our model where the vibration is described by an XY-chain with only nearest-neighbour interaction. As the interaction comes from adjacent strings hindering their motions, different manners of string vibration result in different interaction powers. We also added the discussion above to the revised manuscript. However, the discussion of universality is beyond the scope of this work.

6 . We thank the Referee for the careful proofreading. In the revised manuscript, we have checked the text once again and corrected the mistakes.

Attachment:

Reply.pdf

---

## Round 3 · Author Response

We are thankful to all reviewers for carefully reviewing the paper and providing useful comments and suggestions, which helped us to improve the manuscript.

We appreciate the positive evaluation of both Referees that 'the paper is scientifically sound and well written' and 'the problem is interesting and the results are intriguing'. We also thank their critical comments that are important to understanding the problems in concern. These points have all been properly addressed in the attached reply.

Thanks to the comments of the Referees, we have made substantial improvements to the paper. The revisions made are listed above.

We give a point-by-point response to the comments of all Referees. We believe that the changes made have improved our paper and hope that the current manuscript will be considered suitable for further consideration in SciPost Physics.

---

## Round 3 · List of Changes

1. To address the concern on the possible phases at $J_x<J$ (Comment 1 of Referee 1), we have stressed that the discussion in Section 2, Paragraph 2 only applies to the classical Ising limit $h=0$.
  2. To address the concern on the meaning of the word `vibration' of the string (Comment 3 of Referee 2), we have added reference to related illustration in Section 2, Paragraph 5.
  3. To address the question on the stripe phase (Comment 2 of Referee 2), we have added a related brief discussion to Section 3, Paragraph 2.
  4. To address the concern on the width of incommensurate plateaux (Comment 5 of Referee 2), we have added a sentence to stress the finite size scaling result in Section 3, Paragraph 6.
  5. To address the concern on the ansatz of the effective inter-string interaction (Comment 3 of Referee 1, and Comment 4 of Referee 2), we have added a detailed discussion in Section 4, Paragraph 5 and revised the wording in the Abstract and Conclusion.
  6. To address the concern on the leading order approximation of $B$ (Comment 2 of Referee 1), we have appended a discussion in Section 4, Paragraph 7.

---

## Editorial Decision

resubmitted